# A Case Study on the Variability of Summer Water Properties in the Southeastern Yellow Sea Based on the Hydrological Data from 1995 to 2019

**Lei Wu** [1,2] **and Bin Wang** [1,2,*]

1 Key Laboratory of Marine Hazards Forecasting, Ministry of Natural Resources, Hohai University, Nanjing 210098, China; wl7@hhu.edu.cn
2 College of Oceanography, Hohai University, Nanjing 210098, China
* Correspondence: 20160018@hhu.edu.cn

**Abstract:** The long-term variability of the summer water properties in the southeastern Yellow Sea is described using the hydrological data. The results indicate warming trend in the upper layer and cooling trend in the deeper layer resulting in a strengthen thermocline. While, the mean and the core temperatures in the southeastern Yellow Sea Cold Water Mass both tend to rise slowly, which are previously thought to be fallen. At the deep layer (below 30 m depth), the temperature cooling trend in summer is even stronger than that in winter, which contradict to the stronger surface heat flux in summer. This study proposes that the strengthen thermocline during warming seasons (spring and summer) prevents the heat transferring from surface to the deep layer across isopycnal lines. Furthermore, combined with the metrological data, the strengthen thermocline induces stronger southward tidal residual current at the deep layer, which facilitates the enhanced complementary wind-driven current induced by the southerly monsoon. Thus, the southward tidal induced residual current effectively contributes to the southward motion of the colder water from the northern area. The declined trending of salinity due to the increase of the Changjiang River discharge can be conducive to the strengthen summer thermocline.

**Keywords:** summer water property; hydrological data; tidal residual current; thermocline; southeastern Yellow Sea





## 1. Introduction

The Yellow Sea (YS) is a shallow sea surrounded by the Chinese mainland and the Korean Peninsula. It connects with the Bohai Sea in the northwest, the East China Sea in the south, and the Japan Sea in the southeast through Tsushima Strait, which is a typical continental shelf sea (Figure 1a). The YS plays an important role in maritime trade, fishery resources planning and seabed mineral exploitation for China, North and South Korea and Japan. Traditionally, the YS could be divided into the southern and the northern parts by the line (around 37° N) between the Chengshanton of Shandong Peninsula and the Changsangot of Korean Peninsula. The water depths of the YS are generally shallower than 50 m, including the Subei shoal whose depth is less than 20 m [1]. Moreover, there is a north-south elongated trough with its axis closer to the Korean coasts than to the Chinese coasts in the center of the YS, which is an important conduit of water exchange between the central YS and the nearshore waters along the east coasts of China. The depth of the Yellow Sea Trough gradually decreases from 90 m in the northwest of Cheju Island to 60 m in the north Yellow Sea. The water depth on the west side of the trough changes slowly while that on the east side changes dramatically.

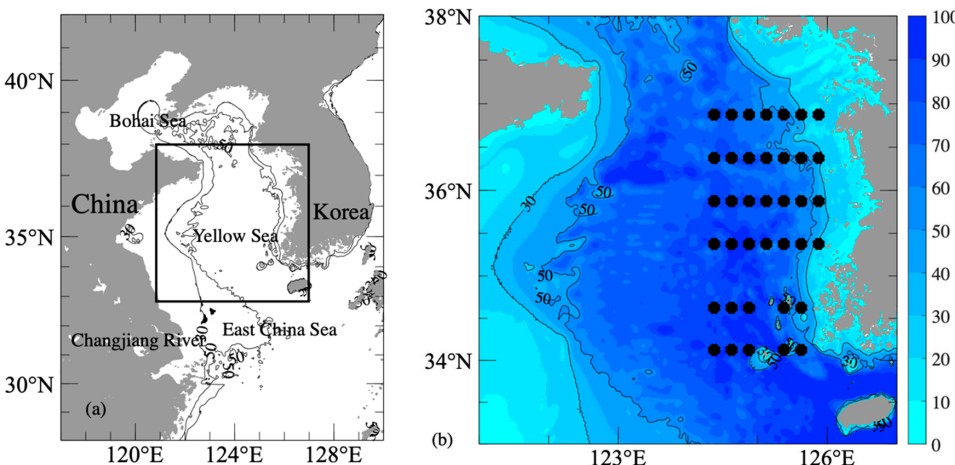

**Figure 1.** Bathymetry map of the Yellow Sea and its surrounding waters (**a**), and the area of interest (**b**). The dot markers represent the locations of survey stations from the Korea Oceanographic Data Center (KODC).

The typical phenomenon of the YS in winter is the Yellow Sea Warm Current (YSWC), which is the only heat source of the Bohai Sea and Yellow Sea in winter. YSWC is characterized by high temperature and high salinity, and has been suggested to be an important and effective mechanism on the distribution and variation of ocean circulation in the YS [2]. As for summer, a typical oceanic phenomenon is the presence of cold water mass in the deep layer of the YS. Due to the existence of thermocline and the relative deep Yellow Sea Trough in the center, the vertical exchange of seawater is inhibited, and the relative cold water, which is the so-called YSCWM, remains under the seasonal thermocline in summer. For many years, researchers have been paying attention to this important oceanic phenomenon, and a large amount of research has been performed.

As early as 1921, Japanese researchers began to investigate the hydrological characteristics in the YS [1]. Uda analyzed the thermohaline distribution and seasonal variation in the North Yellow Sea, discovering the existence of the YSCWM [3]. *He* et al. first studied the formation mechanism of the YSCWM and concluded that it was formed locally during the previous winter [4]. *Guan* found that the temperature of the YSCWM had a relationship with local air temperature during the previous winter [5]. Soon afterwards, many Chinese oceanographers [6–8] adopted the T-S (Temperature and Salinity) curve to analyze water mass properties. On the basis of these investigations, the 10 °C isotherm was considered as the boundary of the YSCWM which has been widely accepted by later scientists [9]. Subsequently, many scientists have also carried out the research on the internal structure of the YSCWM. The YSCWM usually has three cold cores. The northern cold water core, located at around 122°12′ E, 38°14′ N in most years, is called the Northern Yellow Sea Cold Water Mass (NYSCWM). Another two cold cores usually locate between 35° N and 36° N in the southern YS, which are called the eastern and western Southern Yellow Sea Cold Water Mass (E-SYSCWM, W-SYSCWM), respectively [10–12].

Because the lack of the continuous observed data and the deviation of the satellite remote sensing data in China's coastal waters, little research has been performed on the long-term changes of the YSCWM. Yet, the long-term variation of the SYSCWM plays an important role in explanation of yield of the demersal fishes [13]. Since the YSCWM is considered to be highly conservative, its long-term signals are essential to understand climatological evolution of YS [14]. Recently, with accumulation of historical data, many investigations have described the long-term changes of the cold water masses. *Park* et al. conducted Empirical Orthogonal Function (EOF) and Singular Value Decomposition (SVD) analyzing the hydrological data from the Korea Oceanographic Data Center (KODC) and described the temperature variability of the east SYSCWM [14]. They reported that changes in temperature of the eastern part of the SYSCWM are highly related to changes of atmospheric forcing during the warming process. *Yang* et al. used the same data

and suggested that the sea surface temperature (SST) in the previous winter determines the southern edge of the SYSCWM the following summer [15]. *Park* et al. pointed out that the long-term SST warming trends are closely related to the increasing trend of stratification, and its amplitudes are related to the Arctic Oscillation (AO) signals. The authors also mentioned that the temperature of W-SYSCWM was increasing whereas that was decreasing in the E-SYSCWM [16]. *Li* et al. used the section data of KODC and China standard section survey data to discuss the long-term temperature variation of the SYSCWM [17]. The authors indicated that the W-SYSCWM was warmer, which was related to the warm water intrusion of the Yellow Sea Warm Current, the winter meridional wind, the winter local air temperature and the warning process during spring season. Moreover, the stronger thermocline and weakened heat input were supposed to be two main causes of the cooling trend of the E-SYSCWM. Their study only focused on the section approaching 36° N. Although the YSCWM was considered to be quite conservative, there are already a number of studies that have mentioned that the SYSCWM behaves southward seasonal migration, which could transport the colder water of the northern part of YS to the south [18,19]. Furthermore, the tide-induced residual currents under baroclinic conditions, southerly wind and strong surface solar radiation were proposed to be the controlling factors of the SYSCWM migration [19]. However, the long-term variations of the SYSCWM migration were not taken into account in *Li* et al. [17]. The characteristics and variations of main body of the E-SYSCWM and its dynamic mechanism remains unclear.

For the E-SYSCWM, the long-term continuous observations provide the possibility of investigating its long-term trend as a whole. Thus, this study aimed to analyze the temperature and salinity properties of the main body of the E-SYSCWM by using the data of KODC and to explore the dynamics. In Section 2, various datasets used in this study are introduced. Then, we describe the long-term variations of the temperature and salinity in Section 3. In Section 4, the dynamics of the long-term variations of the E-SYSCWM are proposed. The paper ends up with conclusions in Section 5.

## 2. Material and Data

### 2.1. Hydrological Data of KODC

In order to investigate the long-term trend of temperature and salinity of the E-SYSCWM, we used the CTD (Conductivity, Temperature, Depth) data of the Korea Oceanographic Data Center (KODC), which is bimonthly observation data (February, April, June, August, October and December) that have been routinely collected since 1960.

We select the sections (Figure 1b) within the limits of 34° N–37° N, 124°30′ E–127° E as the main body of the E-SYSCWM. The study period was from 1995 to 2019. The original vertical levels of the data were 0, 10, 20, 30, 50, and 75 m, and the horizontal resolution is was ~0.25°. Linear interpolation was trimmed on the original data for comparison. The resolution of final interpolation matrix was 0.1° in horizontal direction and 5 m in the vertical direction. In order to filter out the interannual signals, we used the five-years running mean value, the following data were also treated in this way.

As is well-known, the volume of the SYSCWM has frequently been defined by the 8 °C or 10 °C isotherm in previous studies. Thus, the isotherms of 8 °C and 10 °C at different depths in each summer were checked. We found that the isotherm of 10 °C was closer to the maximum gradient isoline of temperature (not shown). Thus, the isotherm of 10 °C is selected to define the boundary of the SYSCWM, which was consistent with the commonly used definition method of the SYSCWM [9].

### 2.2. Wind Stress of ECMWF Reanalysis v5 (ERA5)

The monthly data of ERA5, the fifth generation ECMWF (European Centre for Medium-Range Weather Forecasts) atmospheric reanalysis of the global climate, was used to analyze the long-term change of the wind stress from 1995 to 2019. ERA5 provides hourly estimates of a large number of atmospheric, land and oceanic climate variables. The data cover the Earth on a 30 km grid and resolve the atmosphere using 137 levels from the surface

up to a height of 80 km. ERA5 combines vast amounts of historical observations into global estimates using advanced modelling and data assimilation systems. *Dee* et al. described the ERA-Interim reanalysis dataset in detail. The horizontal resolution of the data was 0.25° [20].

### 2.3. Downward Heat Flux of NCEP Global Ocean Data Assimilation System (GODAS)

We used the Global Ocean Data Assimilation System (GODAS) to describe the long-term variation of heat flux. The GODAS is developed at the National Centers for Environmental Prediction (NCEP) provides and to provide oceanic initial conditions for the global Climate Forecast System (CFS) newly developed at the NCEP [21]. The horizontal resolution of the monthly mean downward heat flux data was 0.333° in meridional and 1.0° in zonal, respectively.

### 2.4. Changjiang River Discharge

The monthly averaged flow of Datong hydrological station, located at 30°46′ N, 117°37′ E, is used to represent the runoff volume of the Changjiang River, which is provided by the Chinese River Sediment Bulletin. The available time period of this data is from 2002 to 2018.

### 3. Results

According to the previous studies, the YSCWM reaches its prosperity in summer. As mentioned above, the KODC data are nearly available nearly every 2 months, we chose the temperature and salinity data in August to present the properties of the E-SYSCWM and analyze its long-term variations.

In terms of the cold core (minimum) temperature in the E-SYSCWM (Figure 2a), it presented a tiny warming trend during the study period. Moreover, the mean temperature also suggested a slight warming process (Figure 2b). The E-SYSCWM became warmer by nearly 0.02 °C from 1995 to 2019. These results are contrary to those of *Li* et al. [17], who only paid attention on the statues along the 36° N section on the basis of the same hydrological data of KODC. In linear regression, a significant relationship means the value of $R^2$ is close to 1 and the *p*-value is less than the default significance level of 0.05. We noticed that the whole E-SYSCWM showed a slight warming trend, but it was not significant. Therefore, we discuss the process in different vertical layers.

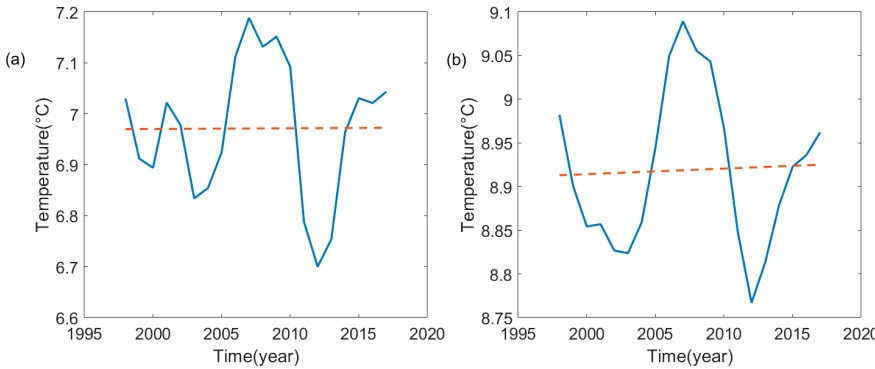

**Figure 2.** Minimal temperature (**a**) and the mean temperature (**b**) the eastern part of the southern Yellow Sea Cold Water Mass (E-SYSCWM) (based on five-years running mean) in August (solid line) and their linear trends (dashed lines). (**a**) $R^2 = 0$ (**b**) $R^2 = 0.0017$.

The horizontal distributions of long-term trend features of temperature and their area-averaged mean values at several depths are shown in Figures 3 and 4. The 10 °C isotherm was used describe the boundary of the E-SYSCWM, as mentioned above, which mostly exists under the depth of 30 m. In the surface layer (Figure 3a), most of the area showed an

obvious warming trend of about 0.08 °C/a on average (Figure 4a), except the southwest corner of the study area. At a depth of 30 m (Figure 3b), the temperature dropped in the most of the study region. It should be pointed out that the main body of the E-SYSCWM, located in the northwestern corner, presented a temperature rising process during the study period. The area-averaged mean temperature at 30 m layer had no significant changes. The cooling trend at the depth of 50 m (Figure 3c) essentially appeared along the eastern edge of the SYSCWM and was less arresting than the upper layer. *Park* et al. indicated that some areas had a cooling trend at 30 m and 50 m, and the cooling trend in the north was weaker than in the south, and there was even an ascending trend in the northwest [16]. The authors' result is consistent with our analysis. The long-term variations of the deep water (70 m) temperature suggest a slightly weaker cooling trend compared with that of 50 m depth (Figures 3d and 4d).

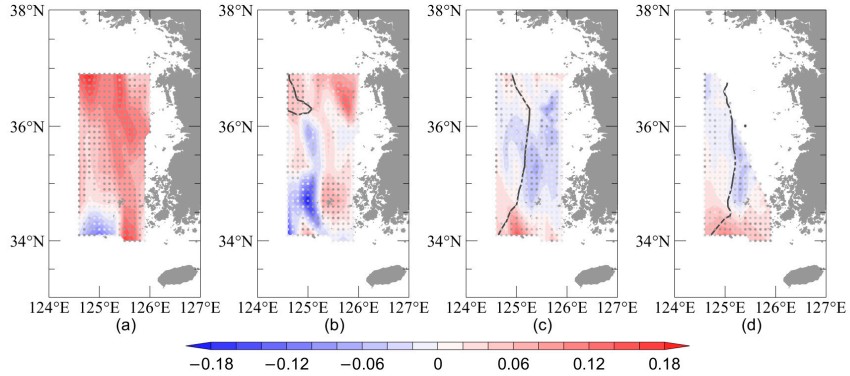

**Figure 3.** Spatial distributions of long-term trends of the temperatures (°C/a) in summer (August) at standard depths (**a**) 10 m, (**b**) 30 m, (**c**) 50 m and (**d**) 70 m during the study period. Solid line is the multi-year averaged edge (10 °C isobath) of the E-SYSCWM. The dots (dark, gray and light) in the figure indicate significance was greater than 95%, greater than 75% but less than 95%, and less than 75%, respectively.

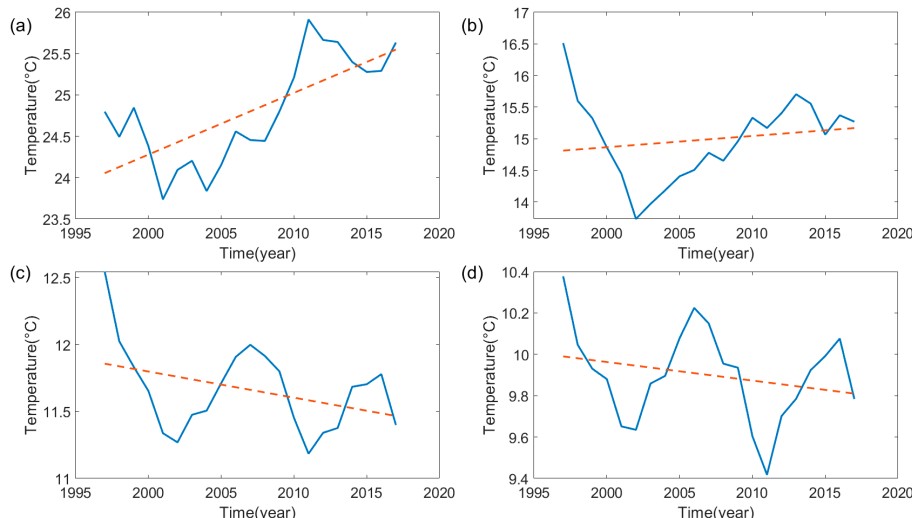

**Figure 4.** The same as Figure 3, but for the area-averaged values. (**a**) $R^2$ = 0.519, $p$ = 0.00023; (**b**) $R^2$ = 0.029, $p$ = 0.46; (**c**) $R^2$ = 0.143, $p$ = 0.091; (**d**) $R^2$ = 0.061, $p$ = 0.27. The trends in subgraphs (**a**) and (**c**) are significant and reliable.

Figure 5 presents the long-term trend features of salinity—the overall trend in all depths was a downward trend. In detail, the downtrend trending in the southern part is smaller, which even turned a negative into a positive in the southern deep layer (not shown). The maximum value of the declining trend at the surface layer was about −0.03 PSU/a,

and the downtrend monotonically weakened with the water depth, which indicts that it was highly related with the freshwater input from the upper layer rather than the intrusion of the YSWC at the deeper layer.

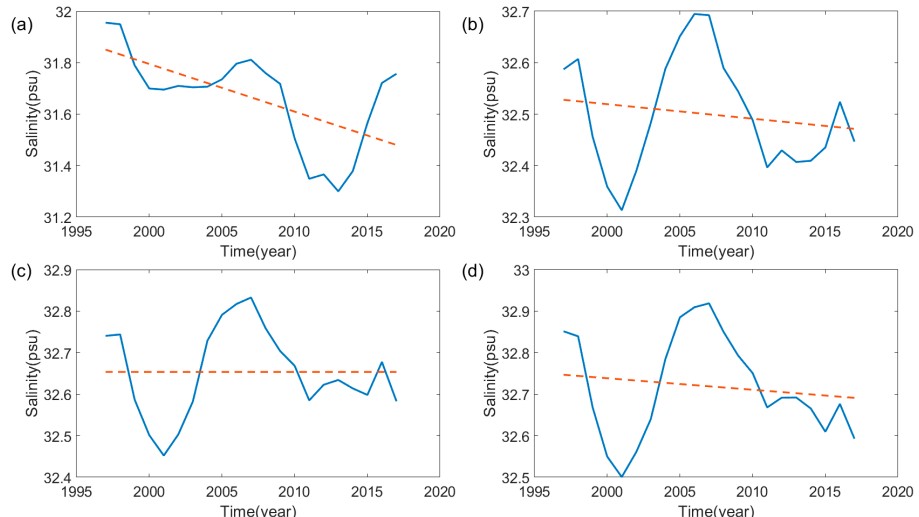

**Figure 5.** Area-averaged long-term (based on five-years running mean) trend of in-situ salinity (PSU/a) in summer (August) at standard depths (**a**) 10 m, (**b**) 30 m, (**c**) 50 m and (**d**) 70 m during the study period. (**a**) $R^2 = 0.384$, $p = 0.0029$ (**b**) $R^2 = 0.0251$, $p = 0.493$ (**c**) $R^2 = 0$, $p = 0.993$ (**d**) $R^2 = 0.019$, $p = 0.54$. The decreasing trend of surface (**a**) salinity was most significant.

With reference to the above analyses, we show the vertical structure of idealized area-averaged temperature and salinity, on the basis of the fitted linear relationships, visually as Figure 6. The warming trend above 30 m depth and the cooling trend in the deeper layer resulted in the strengthening of the thermocline (Figure 6a). Combined with the strengthened halocline (Figure 6b), the pycnocline intensity became stronger from 1995 to 2019, which prevented the mixing across isopycnal lines. In other words, the transferring of heat and fresh water from the upper layer to the deeper layer was gradually blocked.

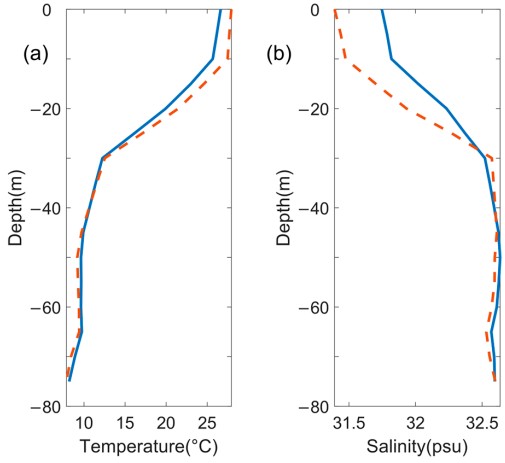

**Figure 6.** The vertical structure of idealized regional average (based on five-years running mean) temperature (**a**) and salinity (**b**) in summer, obtained according to the fitted linear relationship. The solid line is 1995 and dashed line is 2019.

## 4. Dynamics and Discussion

### 4.1. Winter Status

The local water mass in the previous winter is the predecessor of the YSCWM [4]. The temperature of the YS in winter is the initial status of the SYSCWM in summer. Thus, we checked the long-term trend of the winter temperature. Due to the influence of strong mixing, winter temperature of the YS was found to be nearly homogenous in the whole layer. We noticed that the confidence level was not up to standard, which meant the simple linear trend was not as remarkable in winter as in summer. As presented in Figure 7, the overall trend of winter temperature was towards cooling (about −0.04 °C/a), with a slight warming trend (about 0.02 °C/a) in the southwestern and northwestern corners. In other words, the predecessor of the SYSCWM already behaved as a cooling trend during the study period. Moreover, the correlation coefficient between the temperature of the E-SYSCWM and the temperature in the previous winter was 0.47, which was over the 95% significance level of 0.38. In particular, the water in the southern part of the study region behaved more conservative, whose correlation coefficient of two seasons could reach up to 0.62. *Li* et al. focused on the section of 36° N and suggested that there was no significant correlation between winter temperature and summer temperature [17], which indicates that the northern part of the YSCWM might be more strongly affected stronger by the seasonal migration. However, on the whole, the E-SYSCWM remains inactive from the previous winter, which is in accordance with the traditional research [4].

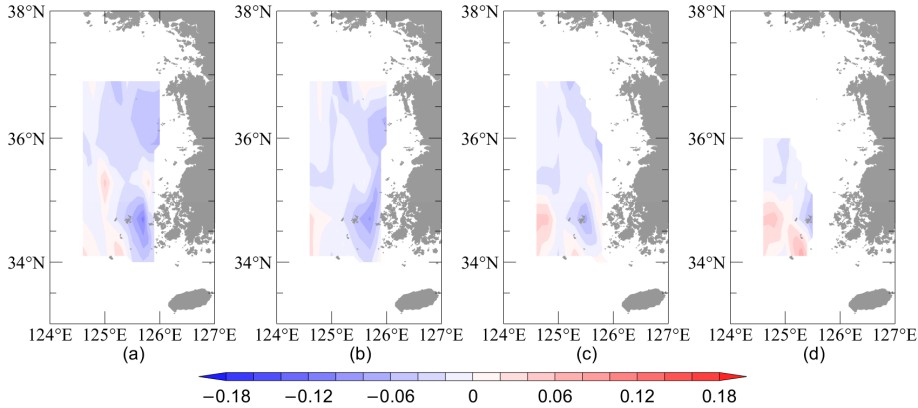

**Figure 7.** Spatial distribution of long-term (based on five-years running mean) trends of the temperature (°C/a) in winter (February) at standard depths (**a**) 10 m, (**b**) 30 m, (**c**) 50 m and (**d**) 70 m during the study period.

The winter meteorological conditions were checked to investigate the mechanism of the long-term variation of the winter temperature in the southeastern YS. The area-avenged surface heat flux in winter suggests the net heat lost from the ocean was becoming larger during the study period (Figure 8). Thus, the YS presents a cooling trend during winter as mentioned above. On the other hand, as shown in Figure 9, the northerly monsoon in winter also became stronger. The area-avenged (117° E–128° E, 28° N–42° N) meridional wind stress enhanced by 8–10% (Figure 9b). According to the previous research [22], the northerly wind drives the compensative up-wind YSWC at the deep layer, which is the only source of the high-temperature and high-salinity water for the YS in wintertime. Consequently, the enhanced YSWC that is invited by the stronger northerly wind could be the reason for the rise trending of the temperature in the southern deep layer during winter. It also explains the slightly salty trend in the southern deep layer as mentioned above.

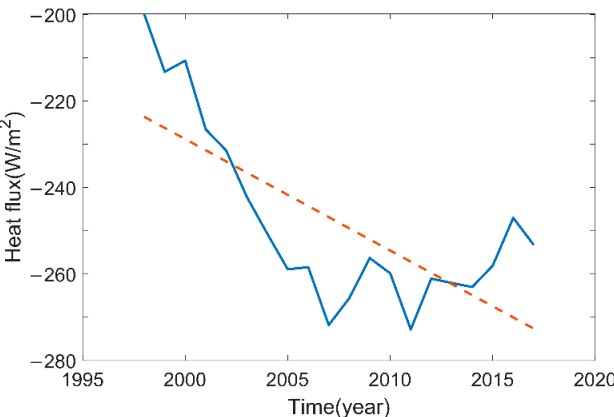

**Figure 8.** Area-avenged (based on five-years running mean) surface heat flux in winter (solid line) with its linear fitting trend (dashed line). $R^2 = 0.529$, $p = 0.0003$. This trend passed the 95% significance test.

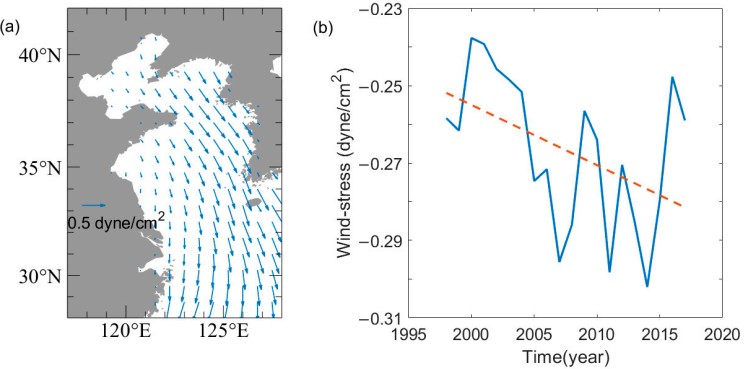

**Figure 9.** Pattern of climatological winter wind stress (February) in the study area (**a**), and (**b**) the area-avenged (based on five-years running mean) meridional wind stress (solid line) with its linear fitting trend (dashed line). (**b**) $R^2 = 0.221$, $p = 0.036$. This trend passed the 95% significance test.

*4.2. Warming Process*

By contrast, the water temperature in summer behaves with a more rapid cooling rate in the deep layer of the YS than in winter (Figures 3 and 7). We examined the long-term variations of the temperature differences between summer (August) and previous winter (February) at standard layers individually, as shown as Figure 10. The surface temperature difference between the two seasons (Figure 10a) showed a clear increase trend ($\approx 0.1\ ^\circ$C/a). However, in the deep layer, it was suggested that the temperature difference between two seasons (Figure 10b–d) was becoming smaller and smaller, which goes against the exacerbated surface warming process. Moreover, it was extraordinary ($-0.18\ ^\circ$C/a) at 30 m depth, where the bottom of the seasonal thermocline is located. Hence, the dynamics of the summer temperature cooling behavior in the deep layer of the YS should be related to the warm-up process during spring and summer.

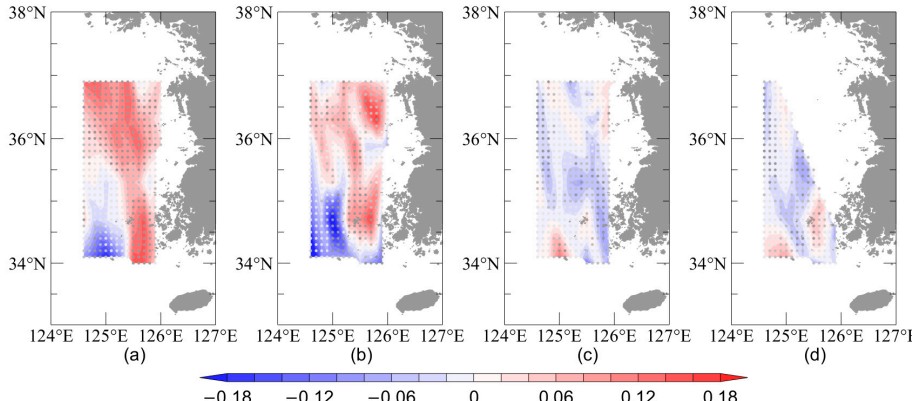

**Figure 10.** Spatial distribution of long-term (based on five-years running mean) trends of the temperatures (°C/a) in the warming process (August minus February) at standard depths (**a**) 10 m, (**b**) 30 m, (**c**) 50 m and (**d**) 70 m during the study period. Shadows pass the significance test with a confidence of 95%. The dots (dark, gray and light) in the figure indicate significance is greater than 95%, greater than 75% but less than 95% and less than 75%, respectively.

The stronger downward net heat flux (Figure 11) during the warming process is conducive to the intensification of the seasonal thermocline, which is an effective heat transferring barrier from the surface to the deep layer. Thus, the warming rate at the bottom of the thermocline (30 m) has been slowed down remarkably. The temperatures between winter and summer are becoming close in the deep layers.

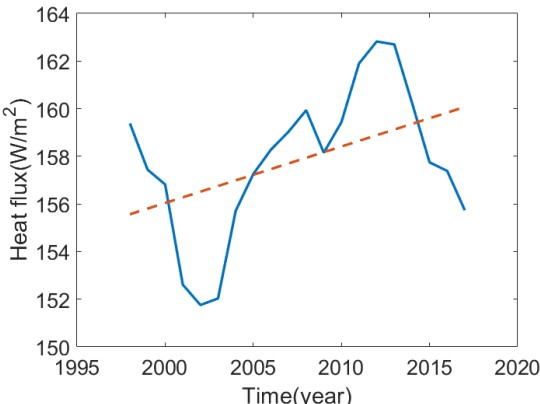

**Figure 11.** Area-avenged (based on five-years running mean) surface heat flux in the warming process (solid line) with its linear fitting trend (dashed line). $R^2 = 0.196$, $p = 0.05$. This trend passed the 95% significance test.

Furthermore, the magnitude of the southward tidal residual current at the deep layer of the YS was found to be highly related with the thermocline and baroclinic condition [19,23], which can drive the colder water from the northern part of YS to the south part and affect the property of the YSCWM. With the advanced intensification of the thermocline during the study period, the stronger southerly tidal residual current was suggested to be also helpful for the cooling summer temperature in the deep layers. During spring and summer, the southerly monsoon prevailed over the YS. Additionally, the southerly (meridional) wind stress presented a weak increasing trend in both spring and summer seasons during the study period (Figure 12). It should be pointed out that the compensative southward current induced by the surface wind forcing is also not ignorable in terms of driving the migration of the YSCWM [19]. The enhanced southerly summer monsoon could reverse an effective contribution to the cooling trend of temperature in the deep layers, namely, the enhanced southward wind-induced current bringing more of the

colder water from the northern area. Further quantitative investigation on the tidal and wind effects on the YSCWM could be conducted on the basis of the numerical modeling.

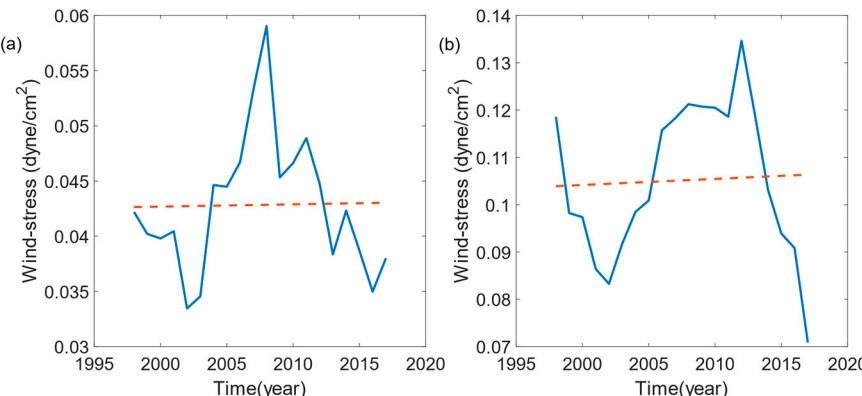

**Figure 12.** Area-avenged (based on five-years running mean) meridional wind stress (solid line) with its linear fitting trend (dashed line) in spring (**a**) and summer (**b**). (**a**) $R^2$ = 0.0004, $p$ = 0.93 (**b**) $R^2$ = 0.002, $p$ = 0.84. There was no significant trend in the long-term variation of wind stress.

In order to investigate the mechanism of the variation of the salinity, we also analyzed the monthly inflow of Datong hydrological station to represent the discharge of Changjiang River (Figure 13). Although the available period of this data was shorter, it still clearly presented the fact that summer runoff of Changjiang River is increasing, which fits the variation features of salinity in the YS, as described in the previous section. In addition, the more freshwater inputs into the sea surface, the larger the vertical salinity gradient generated. It can make a thinly disguised effort to maintain and strengthen the seasonal thermocline.

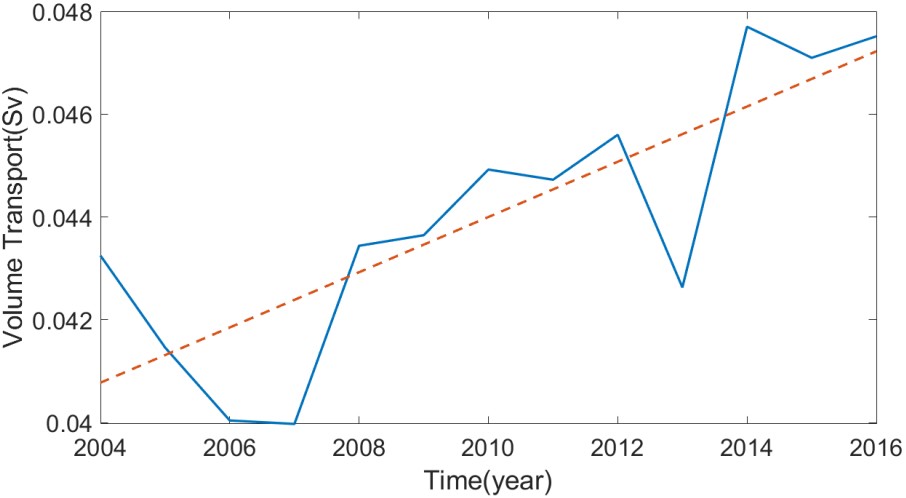

**Figure 13.** Runoff of the Changjiang River (based on five-year running mean) in July and August (solid line) and its linear fitting trend (dashed line). $R^2$ = 0.7, $p$ = 0.0008. This trend passed the 95% significance test.

## 5. Conclusions

The variations of the summer water temperature and salinity in the southeastern Yellow Sea is described using the hydrological data from Korea Oceanographic Data Center. The hydrographic data suggest a warming trend in the upper layer and a cooling trend in the deeper layer resulting in a strengthen summer thermocline in the southeastern Yellow Sea, while the mean temperature and the minimal temperature in the eastern part of the southern Yellow Sea Cold Water Mass both tend to rise slowly. At the deep layer (below

30 m depth), the temperature cooling trend in summer was found to be even stronger than that in winter, which contradicts the stronger surface heat flux in summer. Through combining with the metrological data and the Changjiang River runoff data, we propose the following mechanisms:

1.  The local water mass in previous winter, which is the predecessor of the YSCWM, behaves a slight cooling trend due to more heat lost from the surface. The YSCWM is conservative and highly correlates with the water status in previous winter.
2.  The stronger downward net heat flux and the weaker southerly monsoon strengthen the thermocline during warming seasons (spring and summer), which prevents the heat transferring from the surface to the deep layer across isopycnal lines.
3.  The strengthened thermocline induces a stronger southward tidal residual current at the deep layer, which can be conducive to the migration of the colder water from the northern area. The tidal residual current is combined with the enhanced wind-drive current induced by the southerly monsoon. The quantitative investigations need to be conducted using the numerical modeling.
4.  The increase of the Changjiang River discharge makes an effort to maintain and strengthen the seasonal thermocline.

**Author Contributions:** Conceptualization, L.W. and B.W.; methodology, B.W.; software, L.W.; validation, L.W. and B.W.; formal analysis, B.W.; investigation, B.W.; resources, B.W.; data curation, L.W.; writing—original draft preparation, L.W.; writing—review and editing, B.W.; visualization, L.W.; supervision, B.W.; project administration, B.W.; funding acquisition, B.W. Both authors have read and agreed to the published version of the manuscript.

**Funding:** This work was supported by National Key Research and Development Project (2018YFD0900906), National Natural Science Foundation of China (Project 41706023), the Natural Science Foundation of Jiangsu Province (Grants No. BK20170871). The authors thank the research cruise of the Vessel Sharing Plan of the NSFC and the hard work of the crews and scientists onboard.

**Institutional Review Board Statement:** Not applicable.

**Informed Consent Statement:** Not applicable.

**Data Availability Statement:** The KODC hydrological data could be found from http://www.nifs.go.kr/kodc/eng/eng_soo_summary.kodc. The data of ERA5 could be found from: http://apdrc.soest.hawaii.edu. GODAS data is provided by the NOAA/OAR/ESRLPSD, Boulder, Colorado, USA, from their web site at http://www.esrl.noaa.gov/psd/. The data of Changjiang River Discharge is provided by the Chinese River Sediment Bulletin.

**Conflicts of Interest:** The authors declare no conflict of interest.

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
