# Peer review of "A Case Study on the Variability of Summer Water Properties in the Southeastern Yellow Sea Based on the Hydrological Data from 1995 to 2019"

_water, doi:10.3390/w13010079_

Round 1

Reviewer 1 Report

This manuscript show is presented as a 'Case study' and cannot be considered as an original study.

I've attached the PDF file with several remarks mostly related to minor wording English changes. 

The manuscript uses too many acronyms and difficult to follow sometimes.

The analysis is based on simple trends no other statistical analysis to confirm the trends is presented.

the conclusions are weakly presented with little support of the analysis on the manuscript. 

Author Response

This manuscript show is presented as a 'Case study' and cannot be considered as an original study. I've attached the PDF file with several remarks mostly related to minor wording English changes. The manuscript uses too many acronyms and difficult to follow sometimes.

A: We thank the reviewer for pointing out this defect of our drafting. We have revised the manuscript and have re-written the full title of acronyms you mentioned.

The analysis is based on simple trends no other statistical analysis to confirm the trends is presented. the conclusions are weakly presented with little support of the analysis on the manuscript.

A: In study we focus on the treading of the temperature and salinity in the Yellow Sea. The linear fitting trend is the most straightforward statistical parameter to describe long-term changes in water characteristics. It is true the time-depended variations of the water properties cannot be presented in this study. Further studies on the dynamics of this phenomenon need to be considered.

Reviewer 2 Report

In principle, the paper by Wu and Wang shows the trends in temperature and salinity in the southeastern part of Yellow Sea. Paper is easy to understand, shown results are clear. I think the paper can be accepted for publication after some revision.

I have some questions and remarks to the authors, before I can recommend the paper for publication.

Why is the selected period starting from 1995? I thought you had data from 1960s as said in section 2.1. Please prolong the analysis period.

Why do you use the linear interpolation? There are much better interpolation methods available like optimum analysis/optimum interpolation.

All the figures, where you show linear trend, lack the determination coefficient. What is the p-value i.e. are the trends statistically significant?

Similar work has been done for other seas in the world. For instance, Liblik and Lips (2019, https://www.frontiersin.org/articles/10.3389/feart.2019.00174/full) studied the stratification change in the Baltic Sea using observations. Can you add something similar for the Yellow Sea i.e. how the stratification has changed in the Yellow Sea in general?

Author Response

In principle, the paper by Wu and Wang shows the trends in temperature and salinity in the southeastern part of Yellow Sea. Paper is easy to understand, shown results are clear. I think the paper can be accepted for publication after some revision.

I have some questions and remarks to the authors, before I can recommend the paper for publication.

Why is the selected period starting from 1995? I thought you had data from 1960s as said in section 2.1. Please prolong the analysis period.

A: That is true that the KODC start collected the data since 1960s. However, the consistency and the quality in early year is not so good. And the present study is more interested in modern the water property of the Yellow Sea due to the warming climate. Thus, the study period starts from 1995.

Why do you use the linear interpolation? There are much better interpolation methods available like optimum analysis/optimum interpolation.

A: The spatial resolution of the KODC data is nearly regular. Using different interpolation methods, the results are likely to be the same.

All the figures, where you show linear trend, lack the determination coefficient. What is the p-value i.e. are the trends statistically significant?

A: Thank you very much for your comments. It is worth to pay attentions to the significance of the trend. When we check the statistical significances, we realized that interannual variation and the long term variation are interacted with each other. Thus, we first take a 5-years running mean of the original time series to remove the short-term variations. The modifications and the significance tests have been added in the revised manuscript.

Similar work has been done for other seas in the world. For instance, Liblik and Lips (2019) studied the stratification change in the Baltic Sea using observations. Can you add something similar for the Yellow Sea i.e. how the stratification has changed in the Yellow Sea in general?

A: As method in the manuscript, the observation data in the western Yellow Sea is quite limited, whether in time or space. Thus, the general situation in the whole Yellow Sea is considered to be studied based on numerical models in the next step. Thank you for this comment.

Round 2

Reviewer 1 Report

The authors have improved the manuscript greatly, but I still think it should be published as a case study.

Author Response

Dear reviewer:

Thank you for you comment. We have modified the title to 'A case study on the variability of summer water properties in the southeastern Yellow Sea based on the hydrological data from 1995 to 2019'.

Sincerely,

Bin Wang

Reviewer 2 Report

I can only recommend the paper for the publication in "Water" after minor revision.

In all the figures, where trend line is shown, r-squared and p-values are not shown. Perhaps, it is even worth to add the equation, so one can see the slope.

Author Response

Dear reviewer:

Thank you very much for your comments.
We added the P and R^2 values in the manuscript.
I hope this paper acceptable for “WATER”.

Sincerely,
Bin Wang